# Efficient Length-Generalizable Attention via Causal Retrieval for Long-Context Language Modeling

Xiang Hu [1]   Zhihao Teng [2]   Jun Zhao [3]   Wei Wu [1]   Kewei Tu [2]

## Abstract

Despite the success of Transformers, handling longer contexts remains challenging due to the limited length generalization and quadratic complexity of self-attention. Transformers often require post-training with a larger attention window, significantly increasing computational and memory costs. In this paper, we propose a novel attention mechanism based on dynamic context, **G**rouped **C**ross **A**ttention (GCA), which can generalize to 1000 × the pre-training context length while maintaining the ability to access distant information with a constant attention window size. For a given input sequence, we split it into chunks and use each chunk to retrieve top-$k$ relevant past chunks for subsequent text generation. Specifically, unlike most previous works that use an off-the-shelf retriever, our key innovation allows the retriever to learn how to retrieve past chunks that better minimize the auto-regressive loss of subsequent tokens in an end-to-end manner. Such a mechanism accommodates retrieved chunks with a fixed-size attention window to achieve long-range information access, significantly reducing computational and memory costs during training and inference. Experiments show that GCA-based models achieve near-perfect accuracy in passkey retrieval for 16M context lengths, which is 1000× the training length.

## 1. Introduction

Transformers (Vaswani et al., 2017), serving as the backbone of large language models (LLM), have demonstrated exceptional performance across a wide range of natural language processing tasks (Brown et al., 2020; Achiam et al., 2023;

Touvron et al., 2023; Dubey et al., 2024). While Transformers excel in representational power, they still struggle to process longer contexts as expanding the attention window beyond the pre-trained size may disrupt attention distribution (Wang et al., 2024). Some methods use a sliding window mechanism (Xiao et al., 2024) with a fixed attention field to achieve extrapolation, but they fail to capture long-distance context dependencies outside the attention window. Thus, a main challenge for the attention mechanism lies in how to achieve length generalization but still maintain random access to long-range information. To mitigate the issue, most long-range LMs (Reid et al., 2024) resort to expanding the attention window (Liu et al., 2023) during post-training, significantly increasing training and inference costs due to the quadratic complexity of self-attention.

In this work, inspired by retrieval-based language models (RLM) (Asai et al., 2023), we introduce an efficient attention mechanism that achieves both length generalization and long-range information accessibility. Typically, RLMs (Rubin & Berant, 2024) divide an input sequence into chunks, retrieve relevant ones from the history for the current input, and then integrate the retrieved chunks into the decoder to predict subsequent tokens. By choosing top-$k$ chunks as a "dynamic context", RLMs can maintain the ability to access distant information without expanding the attention window. However, most RLMs (Lewis et al., 2020; Borgeaud et al., 2022) rely on separately pre-trained retrievers, whose retrieved chunks are not necessarily helpful to the causal LMs. Although a straightforward approach is training the retriever end-to-end with the auto-regressive loss, it is rarely explored. The main difficulty is that the relevance scores computed by the retriever do not participate in the next token prediction, thus cannot receive gradient backpropagation from the auto-regressive loss. To overcome the difficulty, we propose **G**rouped **C**ross-**A**ttention (GCA), a novel attention mechanism which integrates chunk-to-chunk retrieval as an inherent ability, thus the retriever can *learn to retrieve* past chunks that most effectively reduce the auto-regressive loss of subsequent tokens, which we refer to as *causal retrieval*.

Specifically, GCA enables the relevance scores to participate in the next token prediction in a differentiable way and can be understood as a chunk-wise analogy of token-wise self-

[1]Ant Group [2]ShanghaiTech University [3]Fudan University. Correspondence to: Wei Wu <congyue.ww@antgroup.com>, Kewei Tu <tukw@shanghaitech.edu.cn>.

*Proceedings of the $42^{nd}$ International Conference on Machine Learning*, Vancouver, Canada. PMLR 267, 2025. Copyright 2025 by the author(s).

attention. In self-attention, considering the next token prediction in causal Transformers, self-attention scores could be viewed as the relevance scores of the current token to past tokens. These scores serve as weights to fuse information gathered from past tokens to predict the next token. Analogously, we divide the input sequence into chunks and use the relevance scores between the *current* and past chunks as weights to fuse past information for the *next* chunk prediction. A detailed comparison between previous works and GCA is depicted in Figure 1. By appending GCA after self-attention in Transformer layers, we introduce **D**ifferentiable **R**etrieval-based **T**ransformers (DRT), enabling pre-training from scratch with context lengths up to 64K. To make pre-training efficient, we sample top-$k$ past chunks according to the relevance scores for each chunk to perform GCA, along with fixed-size sliding window self-attention (Child et al., 2019), achieving linear complexity for the entire input sequence's attention operations. During inference, we offload hidden states of past chunks to CPU memory and reload them when retrieved. It introduces additional memory-swap but largely reduces memory footprint.

To fairly compare GCA with other attention mechanisms, we pre-train all models from scratch and evaluate them on tasks such as long-range language modeling, summarization, and the needle-in-a-haystack (NIAH) tests. The results demonstrate that DRT significantly outperforms all baselines with comparable pre-training costs and much lower inference costs. Notably, in the NIAH test, DRT maintains perfect accuracy on inputs up to 16M tokens. More interestingly, case studies on the arXiv-math dataset validate the causal retrieval ability in DRT, which retrieves lemmas, variants, or functions defined distantly but used in the next chunk. These findings suggest that GCA has the potential to be a fundamental component in long-range LMs. Overall, our main contributions are:

1. We propose a novel attention mechanism called **G**rouped **C**ross-**A**ttention (GCA) and its hardware-aware implementation, which simultaneously achieves length generalization and long-range random-access.[1]

2. Building upon GCA, we introduce **D**ifferentiable **R**etrieval-based **T**ransformers (DRT), which is fast and memory-efficient in both pre-training and inference on long contexts.

3. We conduct comprehensive experiments and demonstrate the promising results of GCA. To the best of our knowledge, GCA is **the first attention mechanism** that can achieve perfect passkey retrieval with 16M context length, 1000 × the pre-training length.

---

[1]The code is released at https://github.com/ant-research/long-context-modeling

## 2. Related works

**Relation to RPT & Landmark Attention.**  There are two long-range LMs closely related to ours. One of them is **R**etrieval-**P**retrained **T**ransformer (RPT) (Rubin & Berant, 2024). The key difference between DRT and RPT is the training approach of the retriever. During data-preparation, for each chunk, RPT picks relevant past chunks by using BM25 (Robertson & Zaragoza, 2009), concatenates them with the current chunk, and evaluates them by a *reference LM* like Pythia 1.4B (Biderman et al., 2023). The past chunks that increase the probability of the next chunk are identified as 'gold chunks' to train RPT's retriever. This process requires that the reference LM be pre-trained on the target dataset beforehand, significantly limiting scalability and flexibility. In contrast, GCA integrate retrieval into the attention mechanism thus can be pre-trained jointly end-to-end without reliance on any external data or models. **L**andmark **A**ttention (LA) (Mohtashami & Jaggi, 2023) is another close work. LA is pre-trained with short contexts but capable of handling long contexts during inference. It addresses long-range language modeling by modifying self-attention KV Cache. During inference, each token, at each layer, selects top-$k$ chunks based on token-to-chunk attention scores and appends their key and value vectors to the current KV cache of self-attention. The token-to-chunk attention scores are trained in an end-to-end manner with a grouped softmax technique. However, it has to perform top-$k$ chunk selection per token, per layer, which incurs significant extra costs during inference. Moreover, it fails to extrapolate on longer context length. Our method combines the chunk-retrieval and grouped softmax ideas, resolving the aforementioned issues while balancing training efficiency and inference performance.

**Attention & Length Generalization.**  The length generalization issue of Transformers stems from the self-attention mechanism, which is usually pre-trained with in-domain position encoding and window size. Consequently, directly inputting longer sequences can lead to out-of-domain issues. Some works mitigate this issue by positional interpolation (bloc97, 2023; Chen et al., 2023; Peng et al., 2024). However, these solutions still generalize only to a limited extent. Even without position encoding, expanding the attention window during inference still fails to extrapolate due to distraction of attention (Wang et al., 2024). To address the discrepancy in attention window between training and inference, some methods (Child et al., 2019; Xiao et al., 2024) employ sliding window attention. However, this sacrifices the model's ability to capture long-range dependencies. Furthermore, even with a fixed attention window, Landmark Attention still encounters length generalization issues. Therefore, developing an attention mechanism that can generalize to various lengths while maintaining long-range information accessibility remains challenging.

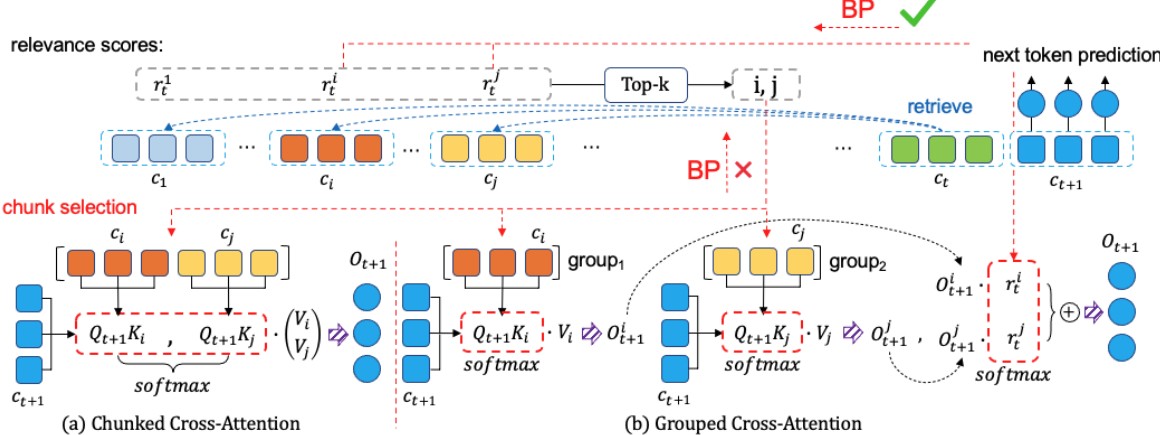

Figure 1: Comparing previous works with GCA. Consider current chunk $c_t$ with its past chunk relevance scores $r_t^k$, where $k \in \{1, \ldots, t-1\}$, $r_t^i$ and $r_t^j$ are the top two. In this example, each chunk contains 3 tokens, whose query, key, and value vectors are denoted as $Q, K, V$. (a) In previous work, information from retrieved chunks is fused into LM decoders via Chunked Cross-Attention, in which relevance scores are merely used for chunk selection. Thus, the loss can not back-propagate to the scores. (b) GCA separately applies Cross-Attention with the two chunks, yielding intermediate outputs $O_{t+1}^i$ and $O_{t+1}^j$. The softmaxed relevance scores serve as weights to fuse these intermediates into LM decoders and thus can receive back-propagation from the loss.

**Long-Range Language Modeling.** Various methods have been proposed to improve long-range language modeling. A straightforward approach is extending the attention window and conducting post-training on a longer context (Bai et al., 2024; Reid et al., 2024; Qwen et al., 2025). However, it comes with a significant additional training cost and still has length generalization issues. One line of research is introducing memorization to Transformers via recurrence. Many works (Dai et al., 2019; Burtsev & Sapunov, 2020; Martins et al., 2022; Hutchins et al., 2022; Munkhdalai et al., 2024) compress past information into fixed-sized vectors. However, these methods often sacrifice the flexibility to attend to arbitrary past tokens. Meanwhile, other works focus on maintaining random-access flexibility of attention. Memorizing Transformers (Wu et al., 2022) appends retrieved past keys and values to the current attention segment via $k$-NN search, but they do not back-propagate gradients to them. CEPE (Yen et al., 2024) parallelly encode long-conext by chunks and fuses them into the decoder via cross-attention. During the training process, the decoder parameters are fixed, and only the encoder is adjusted. Recently, state-space models (Gu & Dao, 2023; Dao & Gu, 2024), RNN models (Beck et al., 2024) and their variants (Nunez et al., 2024) provide new architecture alternatives, with comparable performance to Transformers but much lower cost for inference. A notable distinction of our work is able to achieve perfect past information retrieval for input length $1000 \times$ the pre-training length.

## 3. Methodology

A typical architecture of RLMs (Borgeaud et al., 2022; Yen et al., 2024; Rubin & Berant, 2024) appends Chunked Cross-Attention (CCA) after self-attention to fuse information from retrieved chunks, in which a retriever is merely used to pick relevant chunks. Our approach makes the retriever learnable by replacing CCA with GCA, which represents the key innovation of this work. The novelty of GCA lies in using relevance scores to fuse information from retrieved chunks for LM decoders, enabling the retriever to adaptively learn to select the best past chunks for predicting subsequent tokens, guided by the auto-regressive loss. This section details the model architecture, training, and inference.

### 3.1. Model Architecture

DRT is composed of $N$ Transformer-like layers. Similar to RETRO (Borgeaud et al., 2022), the input sequence of DRT is equally divided into chunks. Formally, given a sequence $\mathbf{x} = [x_1, x_2, ..., x_L]$ with $L$ tokens, we divide the sequence into $\frac{L}{S}$ chunks, where $S$ is the chunk size, denoted as $\{c_1, c_2, ..., c_{L/S}\}$, where $x_i \in c_{\lceil i/S \rceil}$. Similar to Landmark Attention, we insert a special token LMK at the end of each chunk, which summarizes the preceding content via self-attention.

**Forward Pass.** Figure 2 illustrates the forward pass of a token in DRT. DRT layers are bifurcated into upper and lower sections like in RPT (Rubin & Berant, 2024). The key differences are the introduction of GCA and the further division of the upper layers into $G$ groups, enabling learning to retrieve on the fly and adaptive multiple retrievals. The lower layers comprise standard Transformer layers while each upper layer has an additional GCA module after self-attention. In the forward pass, the chunk hidden states output by the lower layers, besides being fed to the upper layers, are also fed into a bi-directional Transformer encoder,

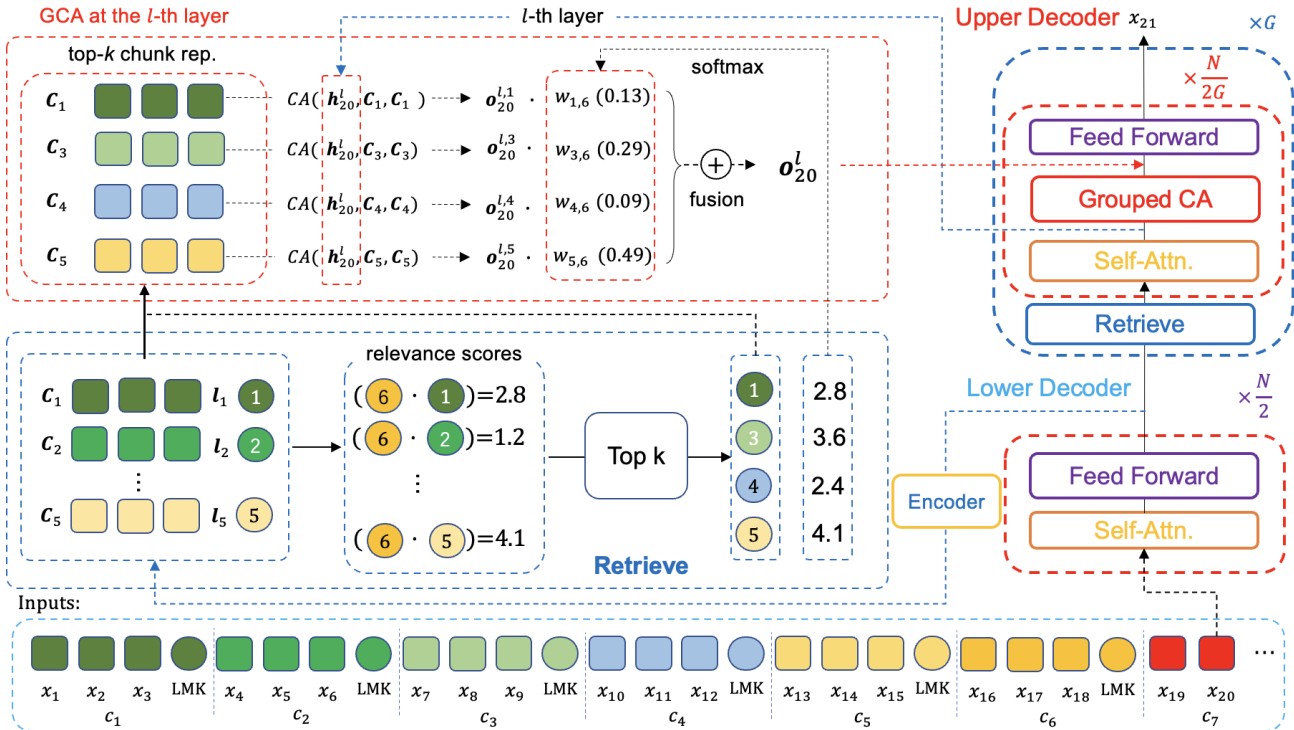

Figure 2: The illustration shows how the current chunk $c_6$ retrieves past chunks for better next token prediction of $x_{20}$ in the next chunk $c_7$. The landmarks's hidden representation is used to compute relevance scores with past chunks, selecting the top four. The hidden states of $x_{20}$ at the $l$-th layer, $\boldsymbol{h}_{20}^l$, perform **C**ross-**A**ttention (CA) with *all* tokens in each *separate* chunk. The chunk-wise CA outputs, $o_{20}^{l,\cdot}$, are fused via weighted sum, whose weights are softmaxed relevance scores.

which further contextualizes the representations with inner-chunk positional encoding, yielding $\boldsymbol{C}_k \in \mathbb{R}^{S \times d}$ and $\boldsymbol{l}_k \in \mathbb{R}^d$ shared across all upper layers, where $d$ is the hidden state dimension. At the $g$-th upper decoder group, chunk $c_t$ retrieves the top-$k$ relevant past chunks for its next chunk:

$$r_t^{g,k} = \frac{(\mathbf{W}_h^g \boldsymbol{h}_t^g)^\top \mathbf{W}_l \boldsymbol{l}_k}{\sqrt{d}}, \quad \mathcal{C}_t^g = \text{Top-k}([r_t^{g,1}, ..., r_t^{g,t-1}]).$$
(1)

Here, $\boldsymbol{h}_t^g \in \mathbb{R}^d$ represents the landmark representation output by the previous decoder layer of the $g$-th group, which accumulates all information from groups 1 to $g-1$. This enables $\boldsymbol{h}_t^g$ to perform multiple retrievals based on retrieved information fused by previous layers. $\mathbf{W}_h^g, \mathbf{W}_l \in \mathbb{R}^{d \times d}$ apply linear transformations on their respective inputs. $r_t^{g,k}$ represents the causal relevance score of $c_k$ to $c_t$. $\mathcal{C}_t^g$ contains the indices of past chunks with top-$k$ relevance scores. The retrieved chunks are shared among the subsequent layers within the same group. The upper layers apply GCA to fuse retrieved information into the decoder.

**Grouped Cross-Attention.** For the $l$-th layer, let $\boldsymbol{H}_{t+1}^l$ and $\hat{\boldsymbol{H}}_{t+1}^l \in \mathbb{R}^{(S+1) \times d}$ denote the batched token representations in the $(t+1)$-th chunk before and after GCA. The function $g(l)$ is used to map a layer index to a group index

and is defined as follows:

$$g(l) = \lceil (l - \frac{N}{2}) / \frac{N}{2G} \rceil.$$

In GCA, we perform Cross-Attention (CA) separately and fuse results via relevance scores:

$$\boldsymbol{O}_{t+1}^{l,k} = \mathbf{CA}(\boldsymbol{H}_{t+1}^l, \boldsymbol{C}_k, \boldsymbol{C}_k), \quad k \in \mathcal{C}_t^{g(l)},$$

$$w_t^{g(l),k} = \frac{\exp(r_t^{g(l),k})}{\sum_{k' \in \mathcal{C}_t^{g(l)}} \exp(r_t^{g(l),k'})},$$

$$\boldsymbol{O}_{t+1}^l = \sum_k w_t^{g(l),k} \boldsymbol{O}_{t+1}^{l,k},$$

$$\hat{\boldsymbol{H}}_{t+1}^l = \text{Norm}(\boldsymbol{H}_{t+1}^l + \boldsymbol{O}_{t+1}^l).$$
(2)

Here $g(l)$ converts the layer index to the group index and $\boldsymbol{C}_k \in \mathbb{R}^{S \times d}$ represents token representations of the $k$-th retrieved chunk. $\boldsymbol{O}_{t+1}^{l,k} \in \mathbb{R}^{(S+1) \times d}$ represents the information that $S + 1$ tokens in chunk $c_{t+1}$ gather from past chunk $c_k$. $w_t^{g(l),k}$ is the normalized relevance score after softmax, serving as the weight of $\boldsymbol{O}_{t+1}^{l,k}$ for information fusion. The final fused results of GCA is $\boldsymbol{O}_{t+1}^l$.

Since $\boldsymbol{C}_k$ is shared across layers, we use the same K, V linear transformations across layers to compact model parameters and reduce memory footprint. For each head $h$, we

have $\mathbf{CA}(\boldsymbol{H}_{t+1}^l, \boldsymbol{C}_k, \boldsymbol{C}_k)_h$ defined as:

$$\text{Softmax}_1\left(\frac{Q_h^l(\boldsymbol{H}_{t+1}^l)K_h(\boldsymbol{C}_k)^T}{\sqrt{d_h}}\right)V_h(\boldsymbol{C}_k) \tag{3}$$

Here, $d_h$ is per-head dimension, and $Q_h^l, K_h, V_h$ are linear transformations per head, where $Q_h^l$ varies across layers and $K_h, V_h$ are layer-shared. We use softmax-off-by-one to allow tokens in the current chunk to ignore any retrieved tokens. The final CA outputs are concatenated vectors from all heads. It is worth noting that GCA is easy to integrate with FlashAttention-2, as detailed in the pseudo-code in Appendix B.

**GCA vs CCA.** A key distinction between GCA and CCA lies in how softmax is applied to cross-attention matrices as shown in Figure 1(a)(b). In CCA, all retrieved chunks are concatenated and softmax is directly applied to the whole attention matrix to fuse token-level information. Notably, relevance scores are entirely excluded from the process. In contrast, GCA applies softmax to each chunk's attention matrix separately. This modification allows information to be gathered from each chunk separately. The softmaxed relevance scores then serve as soft choices (Hu et al., 2021; 2022), participating in the next token prediction thus receiving back-propagated gradients.

### 3.2. Training & Inference

The pre-training objective of DRT is the next token prediction, but there are certain details as discussed below. We also give a detailed time complexity analysis of training.

**Gumbel Top-k sampling.** The core idea of self-supervised retrieval is making candidate chunks compete with each other by softmaxing relevance scores as weights. The weights of the chunks contributing most to the next chunk prediction are enhanced while the weights of the rest are suppressed. To balance exploration and exploitation, we sample chunks based on relevance scores instead of always picking the top-k, enabling highly relevant chunks to be more likely chosen while still considering lower-scoring ones. A simple trick is to add Gumbel noise (Gumbel, 1954) to the raw scores before the top-k operation. Importantly, this noise doesn't affect subsequent operations.

**Time Complexity.** Our approach reduces training complexity by compressing quadratic operations. Vanilla Transformers have a complexity of $\mathcal{O}(NL^2)$ for full self-attention. In DRT, we encode chunks with $S$ tokens into landmark representations, performing chunk-wise full attention to compute relevance scores within $\mathcal{O}(G\frac{L^2}{S^2})$ for $G$ groups of upper layers. By employing sliding-window attention and top-$k$ retrieval, we scale down self-attention and GCA costs to $\mathcal{O}(G\frac{L^2}{S^2} + \frac{N}{2}LKS + NLW)$, where $K$ is the retrieved chunk number and $W$ is the window size, $K, W \ll L$. This

largely reduces the complexity but maintains the random-access flexibility.

**Memory Offloading.** During the inference stage of Transformers, the default memory cost for KV Cache is $\mathcal{O}(NLd)$. To reduce GPU memory usage, we can offload past chunk representations to CPU memory. This results in a spatial complexity of the GPU memory usage $\mathcal{O}(\frac{Ld}{S} + \frac{N}{2}KSd + NWd)$ during inference. Here, $\frac{Ld}{S}$ is the memory footprint of landmark representations, while the remaining terms account for the GCA and sliding-window KV cache. Although each retrieval involves gathering representations from CPU memory and transferring them to the GPU, this operation occurs $G$ times every $S$ tokens. Therefore, the cost of memory exchange from chunk retrievals is minimal.

## 4. Experiments

We compare GCA with other attention mechanisms in language modeling, real-word downstream tasks, and RULER benchmark (Hsieh et al., 2024). Notably, we include the closely related works RPT and Landmark Attention. To ensure fairness, we train all models from scratch with similar configurations.

### 4.1. Experimental Setup

#### 4.1.1. DATASETS

**PG19.** PG19 (Rae et al., 2020) is a language modeling benchmark widely used to evaluate long-range text understanding capabilities of models. It includes a collection of full-length books from Project Gutenberg across a wide range of genres, styles, and topics. The dataset is particularly useful for evaluating models' abilities to capture dependencies and context over long sequences of text.

**ArXiv-math.** ArXiv-math is a corpus consisting of mathematical papers from arXiv. Characterized by a sustained coherence over long distance, the corpus requires models' capability of correctly referring to long-range history information and using them effectively for predictions. We use the preprocessed corpus from (Azerbayev et al., 2023).

#### 4.1.2. MODELS

**DRT**$_{\text{retrieval}\times G}$**.** A DRT consists of 12 Transformer decoder layers, divided into 6 lower and 6 upper layers, with upper layers further split into $G$ groups. The sliding window size is set to $W$=512, the chunk size is set to $S$=64, and 8 chunks are retrieved for GCA, resulting in an attention field of 512 ($8 \times 64$). We implement hardware-aware GCA based on Triton (Tillet et al., 2019). As we employ various parameter settings across different experiments, further details are provided in Appendix A.

| Model | Time | k | attn. win. | PG19↓ valid | PG19↓ test | ArXiv-math↓ valid | ArXiv-math↓ test | PG19↓ valid | PG19↓ test | ArXiv-math↓ valid | ArXiv-math↓ test |
|---|---|---|---|---|---|---|---|---|---|---|---|
| | | | | 128M | | | | 350M | | | |
| Train length=16K, eval length = 2K | | | | | | | | | | | |
| BaseLM | 1× | - | 512 | 15.00 | 14.10 | 3.31 | 3.31 | 12.98 | 12.09 | 3.15 | 3.15 |
| (Sliding window+Alibi) | 1.03× | - | 768 | 14.86 | 13.96 | 3.24 | 3.24 | 12.86 | 11.98 | 3.04 | 3.04 |
| +2 layers | 1.15× | - | 704 | 14.71 | 13.92 | **3.06** | **3.06** | 12.78 | 11.89 | 3.08 | 3.08 |
| RPT$_{contriever}$(our impl.) | 2.5× | 8 | 512 | 14.81 | 13.92 | 3.24 | 3.24 | — | — | — | — |
| Landmark Attn. | 1.5× | 4 | 768 | **14.41** | **13.40** | 3.17 | 3.16 | **12.52** | **11.64** | 2.91 | 2.91 |
| Block Recurrent TFM | 2× | - | 768 | 15.99 | 15.00 | 3.33 | 3.32 | — | — | — | — |
| DRT$_{retrieval×1}$ | 1.22× | 8 | 512 | 14.65 | 13.78 | 3.24 | 3.24 | 12.90 | 12.02 | 3.05 | 3.05 |
| DRT$_{retrieval×2}$ | 1.24× | 8 | 512 | 14.56 | 13.69 | 3.22 | 3.22 | 12.66 | 11.78 | 3.01 | 3.01 |
| Train length=16K, eval length = 16k | | | | | | | | | | | |
| BaseLM | 1× | - | 512 | 14.55 | 13.68 | 3.06 | 3.06 | 12.57 | 11.70 | 2.91 | 2.91 |
| (Sliding window+Alibi) | 1.03× | - | 768 | 14.36 | 13.49 | 2.95 | 2.95 | 12.40 | 11.55 | 2.76 | 2.76 |
| +2 layers | 1.15× | - | 658 | 14.23 | 13.37 | 2.95 | 2.94 | 12.33 | 11.47 | 2.80 | 2.80 |
| RPT$_{contriever}$(our impl.) | 2.5× | 8 | 512 | 14.39 | 13.52 | 2.93 | 2.92 | — | — | — | — |
| Landmark Attn. | 1.5× | 4 | 768 | 14.10 | 13.21 | 3.02 | 3.02 | 12.18 | 11.25 | 2.70 | 2.70 |
| Block Recurrent TFM | 2× | - | 768 | 15.59 | 14.60 | 3.14 | 3.14 | — | — | — | — |
| DRT$_{retrieval×1}$ | 1.22× | 8 | 512 | 14.05 | 13.21 | 2.89 | 2.89 | 12.37 | 11.48 | 2.74 | 2.74 |
| DRT$_{retrieval×2}$ | 1.24× | 8 | 512 | **14.02** | **13.18** | **2.85** | **2.85** | **12.12** | **11.22** | **2.68** | **2.68** |
| BaseLM | 1× | - | 512 | 14.50 | 13.64 | 3.05 | 3.04 | 12.52 | 11.67 | 2.88 | 2.88 |
| (Sliding window+Alibi) | 1.03× | - | 768 | 14.30 | 13.46 | 2.93 | 2.92 | 12.35 | 11.52 | 2.74 | 2.73 |
| +2 layers | 1.15× | - | 658 | 14.18 | 13.34 | 2.93 | 2.92 | 12.29 | 11.43 | 2.78 | 2.77 |
| RPT$_{contriever}$(our impl.) | 2.5× | 8 | 512 | 14.35 | 13.49 | 2.91 | 2.91 | — | — | — | — |
| Landmark Attn. | 1.5× | 4 | 768 | 14.19 | 13.33 | 3.07 | 3.07 | 12.17 | 11.24 | 2.75 | 2.75 |
| Block Recurrent TFM | 2× | - | 768 | 15.61 | 14.56 | 3.13 | 3.12 | — | — | — | — |
| DRT$_{retrieval×1}$ | 1.22× | 8 | 512 | 14.01 | 13.19 | 2.85 | 2.85 | 12.33 | 11.45 | 2.71 | 2.71 |
| DRT$_{retrieval×2}$ | 1.24× | 8 | 512 | **13.98** | **13.16** | **2.81** | **2.81** | **12.08** | **11.19** | **2.65** | **2.65** |
| Ablation studies (Train length=16K, eval length = 16k) | | | | | | | | | | | |
| –w/o Triton | 1.45× | 8 | 512 | — | — | — | — | — | — | — | — |
| –w/o gumbel top-k | 1.22× | 8 | 512 | 14.36 | 13.46 | 2.90 | 2.90 | — | — | — | — |
| –w/ contriever | 2.5× | 8 | 512 | 14.55 | 13.69 | 3.06 | 3.06 | 12.57 | 11.70 | 2.91 | 2.91 |
| –w/ random retriever | 1.22× | 8 | 512 | 14.53 | 13.67 | 3.05 | 3.05 | 12.56 | 11.69 | 2.90 | 2.90 |

Table 1: Perplexity for all datasets. We highlight the best results in **bold** and underline the second best.

**Base LM.** Our base LM is based on the implementation of TinyLlama (Zhang et al., 2024) combined with Flash Attention2 (Dao, 2024) enabling ALiBi (Press et al., 2022) and sliding window attention (Child et al., 2019). We compare models against the baseline across various configurations. One configuration involves 12 layers with a sliding window of 512 tokens, aligning with the DRT sliding window size. Another configuration of 12 layers with a 768-token sliding window ensures the same attention field coverage, as $12 \times 768 = 12 \times 512 + 6 \times 512$(GCA). The strongest baseline, with 14 layers and a 658-token sliding window, has a parameter count comparable to our DRT while maintaining a similar total attention field across all 12 layers, calculated as $658 \times 14 \approx 12 \times 512 + 6 \times 512$.

**Retrieval-Pretrained Transformer (RPT).** Since the official implementation is in JAX and the code for distilling the retriever is not released, we reimplement RPT in PyTorch and replace the retriever with Contriever (Izacard et al., 2022). Elaborations can be found in Appendix E.

**Landmark Attn.** We use the official Llama-like implementation of Landmark Attn. Similar to Base LM, we extend the length of the self-attention range from 512 to 768 to ensure it shares the same attention field as DRT.

**Block-Recurrent TFM.** Since the official implementation of Block-Recurrent Transformer is also based on JAX, we utilized a PyTorch implementation [2] to ensure all baselines are running with the same framework.

[2] https://github.com/lucidrains/block-recurrent-transformer-pytorch

**Ablations.** *w/o Triton*: A naively implemented version of GCA without Triton. *w/o gumbel top-k*: The architecture is exactly the same as DRT, while the only difference lies in the training process by eliminating the gumbel noise when selecting the top-k chunks. *w/ contriever*: We replace the relevance scores in GCA by using an off-the-shelf retriever. *w/ random retriever*: We pick top-$k$ chunks randomly but still keep GCA learnable.

## 4.2. Long-range Language Modeling

In this section, we evaluate DRT against baselines in long-range language modeling on PG19 and arXiv-math, and report their respective perplexities. All models are pre-trained with the same attention field and a 16K context by default, except for baselines that cannot efficiently pre-train on long contexts. To ensure fairness, we adjusted these baselines. Detailed hyper-parameters are provided in Appendix A.

**Results.** From Table 1, we have several observations. **Firstly**, DRT outperforms all baselines where the evaluation length exceeds 16K. While DRT performs retrieval for $G$ times every 64 tokens, LA performs retrieval at every token and in every layer, potentially offering better random access flexibility. However, DRT still surpasses LA on longer inputs. This is likely because LA follows a "train short, test long" approach due to the need for full attention during pre-training, whereas DRT can be directly pre-trained on longer sequences thanks to GCA. DRT can randomly access distant contexts during pre-training with a fixed attention field, allowing it to better utilize long-range information during pre-training with negligible extra training costs. **Secondly**, in terms of parameter efficiency, we compared against Base LM with two additional layers in the decoder stack. It can be observed that on shorter evaluation lengths, the baseline has a slight advantage. However, with a longer context, our model consistently leads. This experiment suggests precisely retrieving semantic knowledge from a long context may be more beneficial for improving the language model than increasing model parameters. **Thirdly**, multiple retrievals yield positive gains with a low marginal cost. It allows upper layers to access more diverse chunks and enables further retrieval based on previous retrieval results. **Finally**, ablation studies show that all the training techniques we add bring positive improvements. In conclusion, the above results fully demonstrate that the GCA module can indeed bring effective gains in modeling long texts, and it is more advantageous compared to other baselines.

### 4.3. Downstream task evaluation & Efficiency Analysis

In this section, we fine-tune all baselines under the same configurations and evaluate them against downstream tasks, including summarization (Nallapati et al., 2016; Narayan

et al., 2018), NIAH tests akin to RULER. The details for the NIAH tests are described in Appendix D. Then we analyze the inference cost, the relationship between training time and context length, the training throughput, and the extrapolation capability of DRT. In the inference cost analysis, We skip RPT because it has a similar cost to DRT. In the extrapolation experiments, we utilize the pre-trained models described in § 4.2 to assess their perplexity on longer inputs.

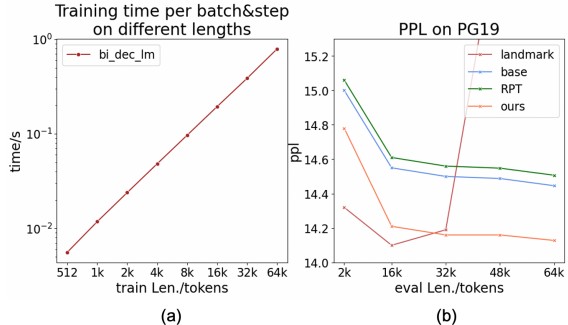

Figure 3: Training speed, extrapolation ability

**Results.** From Table 2, we observe that DRT significantly outperforms all baselines in the summarization tasks, validating its capability to effectively utilize long contexts. Notably, in the single NIAH test, DRT maintains 100% performance even with a context length of up to 16 million tokens, demonstrating its strong length generalization ability in long-context scenarios. Furthermore, in the 2-hop NIAH test, DRT$_{\text{retrieval}\times 2}$ performs comparably to Landmark Attn at context lengths below 16K tokens while successfully extrapolating to longer context lengths beyond 64K tokens. Additionally, DRT$_{\text{retrieval}\times 2}$ significantly outperforms DRT$_{\text{retrieval}\times 1}$, confirming our hypothesis that conducting causal retrieval every $G$-layer contributes to scenarios requiring multiple retrievals.

Table 3 shows DRT significantly outperforms Landmark Attn in terms of memory footprint and inference speed. The main overhead for Landmark Attn arises from modifications to the self-attention KV cache and gathering tensors from memory offloaded to the CPU. In DRT, thanks to the chunk-wise retrieval, we perform retrieval only once every 64 tokens, which is $1/(12 \times 64)$ of the corresponding operation in Landmark Attn.

From Figure 3(a), it can be observed that the total training time increases linearly with the increment of the sequence length. In the extrapolation experiments, both BaseLM and DRT perform well as shown in Figure 3(b). However, Landmark Attn fails to extrapolate at longer eval lengths. While both methods use a fixed attention window and dynamically select past chunks to fill it, why can GCA extrapolate over 1000 times with stable PPL and perfect passkey retrieval, while Landmark Attn fails? We speculate a major reason lies

| Models 128M | Retrieval | Single NIAH↑ | | | | | | | |
|---|---|---|---|---|---|---|---|---|---|
| | | 4K | 8K | 16K | 32K | 64K | 128K | 256K | 16M |
| Base LM (+2 layers) | – | 15.37 | 8.30 | 3.89 | 2.13 | 0.0 | - | - | - |
| Block Recurrent TFM | – | 13.96 | 7.60 | 6.01 | 2.13 | 4.29 | - | - | - |
| RPT$_{\text{Contriever}}$ | fixed | 11.66 | 6.71 | 4.24 | 1.42 | 2.86 | - | - | - |
| Landmark Attn. | adaptive | 99.82 | 97.74 | 97.88 | 96.45 | 0.00 | 0.00 | - | - |
| DRT$_{\text{retrieval}\times1}$ | adaptive | 98.50 | 98.76 | 98.59 | **100.00** | **100.00** | 100.00 | 100.00 | 100.00 |
| DRT$_{\text{retrieval}\times2}$ | adaptive | 99.65 | 99.47 | **99.65** | 99.99 | **100.00** | 100.00 | 100.00 | 100.00 |
| For reference | | | | | | | | | |
| Yarn-base (7B) | – | 100.00 | 100.00 | 99.65 | 99.82 | 100.00 | 42.32 | - | - |
| CEPE (7B) | – | 100.00 | 100.00 | 55.45 | - | - | - | - | - |

| Models 128M | XSum↑ | | | CNN/DailyMail↑ | | | 2-hop NIAH ↑ | | | | | |
|---|---|---|---|---|---|---|---|---|---|---|---|---|
| | R-1 | R-2 | R-L | R-1 | R-2 | R-L | 1K | 4K | 16K | 64K | 256K | 4M |
| BaseLM | 29.43 | 8.04 | 23.26 | 32.38 | 12.57 | 22.61 | 3.60 | 1.15 | 0.71 | 0.0 | 0.0 | - |
| +2 layers | 29.74 | 8.26 | 23.48 | 34.14 | 14.09 | 23.60 | 16.60 | 5.83 | 1.06 | 0.0 | 0.0 | - |
| Landmark Attn. | 27.98 | 6.96 | 21.99 | 34.06 | 13.80 | 23.77 | **90.82** | **88.35** | **86.41** | 0.0 | 0.0 | - |
| DRT$_{\text{retrieval}\times1}$ | 30.30 | 8.59 | 23.92 | 36.27 | 15.88 | 25.08 | 41.07 | 33.39 | 39.93 | 38.57 | 35.29 | 34.29 |
| DRT$_{\text{retrieval}\times2}$ | **30.39** | **8.64** | **23.98** | **36.39** | **15.96** | **25.15** | 88.52 | 84.45 | 86.21 | **81.43** | **94.11** | **79.41** |

Table 2: The performances of various models on summarization and tasks akin to RULER.

| | Prompt #tokens | Generated #tokens | w/ cpu offload | | w/o cpu offload | |
|---|---|---|---|---|---|---|
| | | | time/token↓ | mem. cost↓ | time/token↓ | mem. cost↓ |
| Landmark Attn. / Base LM | 16K | 128 | 160.9× | 1.98× | 4.16× | 32× |
| | 48K | 128 | 163.1× | 2.98× | 4.25× | 96× |
| Block Recurrent TFM / Base LM | 16K | 128 | - | - | 2.85× | 2× |
| | 48K | 128 | - | - | 2.85× | 2× |
| DRT$_{\text{retrieval}\times1}$ / Base LM | 16K | 128 | 1.25× | 1.54× | 1.06× | 4.08× |
| | 48K | 128 | 1.27× | 1.62× | 1.08× | 9.41× |

Table 3: The inference time per token and memory footprint ratio compared to the Base LM (12 layers with a 512 sliding window), with lower values indicating better performance.

| Model | Throughput #params.=350M, ctx-len=32K | Memory | Throughput #params.=760M, ctx-len=16K | Memory | Throughput #params.=1.5B, ctx-len=8K | Memory | Throughput #params.=3B, ctx-len=4K | Memory |
|---|---|---|---|---|---|---|---|---|
| Transformer$_{\text{SW}}$ | $2.91 \times 10^5$ | 50G | $1.31 \times 10^5$ | 50G | $6.56 \times 10^4$ | 50G | $3.41 \times 10^4$ | 72G |
| Transformer$_{\text{full\_attn}}$ | $4.44 \times 10^4$ | 50G | $4.00 \times 10^4$ | 50G | $3.50 \times 10^4$ | 72G | $2.85 \times 10^4$ | 72G |
| BRT | $7.94 \times 10^4$ | 59G | $5.70 \times 10^4$ | 56G | $4.02 \times 10^4$ | 59G | — | OOM |
| RPT$_{\text{Our\_impl}}$ | $1.13 \times 10^4$ | 57G | $5.17 \times 10^4$ | 59G | $2.73 \times 10^4$ | 64G | $1.45 \times 10^4$ | 80G |
| DRT$_{\text{retrieval}\times1}$ | $2.32 \times 10^5$ | 56G | $1.06 \times 10^5$ | 59G | $5.52 \times 10^4$ | 63G | $2.93 \times 10^4$ | 80G |
| DRT$_{\text{retrieval}\times2}$ | $2.28 \times 10^5$ | 57G | $1.05 \times 10^5$ | 59G | $5.52 \times 10^4$ | 63G | $2.93 \times 10^4$ | 81G |

Table 4: Training throughput of models with parameters ranging from 350M to 3B.

in the different objectives for optimizing retrievers. Landmark Attn optimizes for predicting the next token based on retrieved chunks, aiming for maximizing $P(x_t|\mathcal{R}(x_{<t}))$ where $\mathcal{R}(x_{<t})$ denotes retrieved chunks. In contrast, GCA retrieves past chunks for the next chunk prediction, aiming to maximize $P(x_{t:t+S}|\mathcal{R}(x_{<t}))$. A single token offers limited information, which may result in false positive retrievals thus leading to inefficacy in larger retrieval spaces. While chunks provide richer semantic information, enabling generalization to larger search spaces.

### 4.4. Case Studies

By analyzing DRT's retrieval results on the arXiv-math dataset, we find some intriguing cases. A case is given

in Figure 4. When retrieving past chunks, the results not only include the definition of prepositions referenced in the current chunk but also lemmas to be used in the next chunk. This validates the idea of causal retrieval, allowing us to not only retrieve semantically similar content but also information that better predicts the next chunk. More case studies can be found in Appendix C.

## 5. Conclusion & Future works

In this study, we introduce a length-generalizable attention mechanism that maintains efficient long-range information access. Notably, it achieves perfect accuracy in passkey retrieval on 16M contexts—a first in the field. The core in-

*...     We start first with the following* lemma which is useful to establish the Quillen equivalence. \begin{lemma} \label{lem-reflect-equiv} Let **M** be a symmetric monoidal model category that is combinatorial and left proper. Assume that the transferred (natural) model structure on *com* exists. *Let* $\sigma : \mathbf{C} \to \mathbf{D}$ *be a morphism between usual commutative...*

*...     weak equivalence between co-Segal categories is just a level-wise weak equivalence.* \begin{prop} \label{prop-eta-kx-loc-equiv} For any $\mathbf{F} \in coms$, the canonical map $\mathbf{F} \to |\mathbf{F}|$ is an equivalence in *comsepc* i.e, it's a $kb(I)$-local equivalence in *comsep (whence in comse).* \end{prop}*...*

*...     Thanks to* **Proposition \ref{prop-eta-kx-loc-equiv}** *,* **we know that** $\eta : \mathbf{F} \to |\mathbf{F}|$ **is always a** $kb(I)$**-local equivalence.** **Then by** 3**-for-**2 *we see that* $\sigma$ *is a* $kb(I)$*-local equivalence if and only if* $ol(\sigma)$ *is.* *Now thanks to* Lemma \ref{lem-reflect-equiv} *we know that ...*

Figure 4: In the case above, retrieved 2nd best chunk describes a proposition which appears in the **current chunk**. retrieved best chunk and retrieved 3rd best chunk are adjacent chunks which introduce the lemma used in the next chunk.

novation lies in the Grouped Cross-Attention (GCA), which makes relevance scores learnable by using them to fuse information retrieved by the current chunk for next chunk prediction. Combined with Gumbel top-k sampling, this approach enables the pre-training of LMs efficiently on context lengths extending up to 64K tokens.

In future work, we will explore self-supervised causal retrieval from vast amounts of tokens outside the context. Meanwhile, we will combine structured representations (Hu et al., 2024b;a) to achieve multi-granular retrieval.

## Impact Statement

This paper presents a novel attention mechanism that aims to address the length generalization issue of LLMs. There are many potential societal consequences of our work associated with LLMs. Beyond LLMs, we feel no other consequences must be highlighted here.

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

# A. Hyper-parameters

**Long-Range Language Modeling.** We employ a Llama-like architecture (Touvron et al., 2023) featuring a 12-layer, decoder-only transformer with 12 heads per layer (64 dimensions each), an embedding dimension of 768, and an FFN size of 2048. Training utilizes the AdamW optimizer (Loshchilov & Hutter, 2019) with $\beta_1 = 0.9$ and $\beta_2 = 0.95$, and a weight decay factor of 0.001. We used base learning rate $2 \times 10^{-3}$ for all our experiments with a warmup stage that was 2% of the whole training and applied a cosine scheduler with final learning rate being $4 \times 10^{-4}$. We used GPT-2's (Radford et al., 2019) tokenizer. We used mixed-precision training with bfloat16 over at 8 Nvidia A100 GPUs. We train all models with an effective batch size of $2^{19}$ tokens for 60K steps resulting in a total training budget of 32.2 billion tokens. We train Base LM, RPT and DRT on each dataset with a context length of 16K tokens. Due to Landmark Attention doesn't support sliding-window attention, the model is pre-trianed with full self-attention with a context length of 768. Due to Block Recurrent Transformer cannot be fully paralleled, which takes $5\times$ wall-clock training time with 16K context length, we pre-train it with a context length of 4K.

# B. Hardware-aware GCA psuedo-code

---

**Algorithm 1** FLASHGCA forward pass

---

**Require:** Matrices $\mathbf{Q} \in \mathbb{R}^{N_q \times d}$, $\mathbf{K}, \mathbf{V} \in \mathbb{R}^{K \times N_{kv} \times d}$ in HBM, vector $\boldsymbol{w} \in \mathbb{R}^k$ in HBM, block sizes $B_c$, $B_r$.

1: Divide $\mathbf{Q}$ into $T_r = \left\lceil \frac{N_q}{B_r} \right\rceil$ blocks $\mathbf{Q}_1, \ldots, \mathbf{Q}_{T_r}$ of size $B_r \times d$ each, and divide $\mathbf{K}, \mathbf{V}$ in to $K \times T_c$ blocks where $T_c = \left\lceil \frac{N_{kv}}{B_c} \right\rceil$
    $\mathbf{K}_{1,1}, \ldots, \mathbf{K}_{K,T_c}$ and $\mathbf{V}_{1,1}, \ldots, \mathbf{V}_{K,T_c}$, of size $B_c \times d$ each.

2: Divide the output $\mathbf{O} \in \mathbb{R}^{N_q \times d}$ into $T_r$ blocks $\mathbf{O}_i, \ldots, \mathbf{O}_{T_r}$ of size $B_r \times d$ each, and divide the logsumexp $L \in \mathbb{R}^{N_q \times K}$ into $T_r \times K$
    blocks $L_{1,1}, \ldots, L_{T_r,K}$ of size $B_r$ each.

3: Divide the output $\mathbf{O}' \in \mathbb{R}^{K \times N_q \times d}$ into $T_r$ blocks $\mathbf{O}_{1,1}, \ldots, \mathbf{O}_{K,T_r}$ of size $K \times B_r \times d$ each.

4: **for** $1 \leq i \leq T_r$ **do**

5:     Load $\mathbf{Q}_i$ from HBM to on-chip SRAM.

6:     Load $\boldsymbol{w}_k$ from HBM to on-chip SRAM.

7:     **for** $1 \leq k \leq K$ **do**

8:         On chip, initialize $\mathbf{O}_i^{(0)} = (0)_{B_r \times d} \in \mathbb{R}^{B_r \times d}, \ell_i^{(0)} = (0)_{B_r} \in \mathbb{R}^{B_r}, m_i^{(0)} = (-\infty)_{B_r} \in \mathbb{R}^{B_r}$.

9:         **for** $1 \leq j \leq T_c$ **do**

10:             Load $\mathbf{K}_{k,j}, \mathbf{V}_{k,j}$ from HBM to on-chip SRAM.

11:             On chip, compute $\mathbf{S}_i^{(j)} = \mathbf{Q}_i \mathbf{K}_{k,j}^T \in \mathbb{R}^{B_r \times B_c}$.

12:             On chip, compute $m_i^{(j)} = \max(m_i^{(j-1)}, \mathrm{rowmax}(\mathbf{S}_i^{(j)})) \in \mathbb{R}^{B_r}$, $\tilde{\mathbf{P}}_i^{(j)} = \exp(\mathbf{S}_i^{(j)} - m_i^{(j)}) \in \mathbb{R}^{B_r \times B_c}$ (pointwise),
    $\ell_i^{(j)} = e^{m_i^{j-1} - m_i^{(j)}} \ell_i^{(j-1)} + \mathrm{rowsum}(\tilde{\mathbf{P}}_i^{(j)}) \in \mathbb{R}^{B_r}$.

13:             On chip, compute $\mathbf{O}_i^{(j)} = \mathrm{diag}(e^{m_i^{(j-1)} - m_i^{(j)}})^{-1} \mathbf{O}_i^{(j-1)} + \tilde{\mathbf{P}}_i^{(j)} \mathbf{V}_{k,j}$.

14:         **end for**

15:         On chip, compute $\mathbf{O}'_{i,k} = \mathrm{diag}(\ell_i^{(T_c)})^{-1} \mathbf{O}_i^{(T_c)}$.

16:         On chip, compute $\mathbf{O}_i \leftarrow \mathbf{O}_i + \boldsymbol{w}_k \mathbf{O}'_{i,k}$.

17:         Write $O_{i,k}$ to HBM.

18:         On chip, compute $L_{i,k} = m_i^{(T_c)} + \log(\ell_i^{(T_c)})$.

19:         Write $L_{i,k}$ to HBM.

20:     **end for**

21:     Write $\mathbf{O}_i$ to HBM as the $i$-th block of $\mathbf{O}$.

22: **end for**

23: Return the output $\mathbf{O}$ and the logsumexp $L$.

---

---

**Algorithm 2** FLASHGCA Backward Pass

---

**Require:** Matrices $\mathbf{Q}, \mathbf{O}, \mathbf{dO} \in \mathbb{R}^{N_q \times d}, \mathbf{K}, \mathbf{V} \in \mathbb{R}^{K \times N_{kv} \times d}, L \in \mathbb{R}^{N_q \times K}, \mathbf{O}' \in \mathbb{R}^{K \times N_q \times d}$ in HBM, vector $\boldsymbol{w} \in \mathbb{R}^K$ in HBM, block sizes $B_c, B_r$.

1: Divide $\mathbf{Q}$ into $T_r = \left\lceil \frac{N}{B_r} \right\rceil$ blocks $\mathbf{Q}_1, \ldots, \mathbf{Q}_{T_r}$ of size $B_r \times d$ each, and divide $\mathbf{K}, \mathbf{V}$ in to $K \times T_c$, where $T_c = \left\lceil \frac{N}{B_c} \right\rceil$ blocks $\mathbf{K}_{1,1}, \ldots, \mathbf{K}_{K,T_c}$ and $\mathbf{V}_{1,1}, \ldots, \mathbf{V}_{K,T_c}$, of size $B_c \times d$ each.

2: Divide $\mathbf{O}$ into $T_r$ blocks $\mathbf{O}_i, \ldots, \mathbf{O}_{T_r}$ of size $B_r \times d$ each, divide $\mathbf{dO}$ into $T_r$ blocks $\mathbf{dO}_i, \ldots, \mathbf{dO}_{T_r}$ of size $B_r \times d$ each, and divide $L$ into $T_r \times K$ blocks $L_{1,1}, \ldots, L_{T_r,K}$ of size $B_r$ each.

3: Initialize $\mathbf{dQ} = (0)_{N_q \times d}$ in HBM and divide it into $T_r$ blocks $\mathbf{dQ}_1, \ldots, \mathbf{dQ}_{T_r}$ of size $B_r \times d$ each. Divide $\mathbf{dK}, \mathbf{dV} \in \mathbb{R}^{K \times N_{kv} \times d}$ in to $K \times T_c$ blocks $\mathbf{dK}_{1,1}, \ldots, \mathbf{dK}_{K,T_c}$ and $\mathbf{dV}_{1,1}, \ldots, \mathbf{dV}_{K,T_c}$, of size $B_c \times d$ each. Initialize $\boldsymbol{dW} = (0)_{T_r \times K}$ in HBM.

4: Compute $D = \text{rowsum}(\mathbf{dO} \circ \mathbf{O}') \in \mathbb{R}^{N_q \times K}$ (pointwise multiply), write $D$ to HBM and divide it into $T_r$ blocks $D_1, \ldots, D_{T_r}$ of size $B_r$ each.

5: **for** $1 \le k \le K$ **do**

6:     Load $\boldsymbol{w}_k$ from HBM to on-chip SRAM.

7:     **for** $1 \le j \le T_c$ **do**

8:         Load $\mathbf{K}_{k,j}, \mathbf{V}_{k,j}$ from HBM to on-chip SRAM.

9:         Initialize $\mathbf{dK}_{k,j} = (0)_{B_c \times d}, \mathbf{dV}_{k,j} = (0)_{B_c \times d}, \boldsymbol{dW}_{k,j} = (0)$ on SRAM.

10:         **for** $1 \le i \le T_r$ **do**

11:             Load $\mathbf{Q}_i, \mathbf{dO}_i, \mathbf{dQ}_i, D_i$ from HBM to on-chip SRAM.

12:             Load $L_{i,k}$ from HBM to on-chip SRAM.

13:             On chip, compute $\mathbf{S}_i^{(j)} = \mathbf{Q}_i \mathbf{K}_{k,j}^T \in \mathbb{R}^{B_r \times B_c}$.

14:             On chip, compute $\mathbf{P}_i^{(j)} = \exp(\mathbf{S}_{ij} - L_{i,k}) \in \mathbb{R}^{B_r \times B_c}$.

15:             On chip, compute $\mathbf{dV}_{k,j} \leftarrow \mathbf{dV}_{k,j} + (\boldsymbol{w}_k \mathbf{P}_i^{(j)})^\top \mathbf{dO}_i \in \mathbb{R}^{B_c \times d}$.

16:             On chip, compute $\mathbf{dP}_i^{(j)} = \mathbf{dO}_i \mathbf{V}_j^\top \in \mathbb{R}^{B_r \times B_c}$.

17:             On chip, compute $\boldsymbol{dW}_{i,k} = \text{rowsum}(\mathbf{P}_i^{(j)} \circ \mathbf{dP}_i^{(j)})$.

18:             On chip, compute $\mathbf{dS}_i^{(j)} = \boldsymbol{w}_k \mathbf{P}_i^{(j)} \circ (\mathbf{dP}_i^{(j)} - D_{i,k}) \in \mathbb{R}^{B_r \times B_c}$.

19:             Write $\boldsymbol{dW}_{i,k}$ to HBM.

20:             Load $\mathbf{dQ}_i$ from HBM to SRAM, then on chip, update $\mathbf{dQ}_i \leftarrow \mathbf{dQ}_i + \mathbf{dS}_i^{(j)} \mathbf{K}_j \in \mathbb{R}^{B_r \times d}$, and write back to HBM.

21:             On chip, compute $\mathbf{dK}_{k,j} \leftarrow \mathbf{dK}_{k,j} + \mathbf{dS}_i^{(k,j)^\top} \mathbf{Q}_i \in \mathbb{R}^{B_c \times d}$.

22:         **end for**

23:         Write $\mathbf{dK}_{k,j}, \mathbf{dV}_{k,j}$ to HBM.

24:     **end for**

25: **end for**

26: $\boldsymbol{dW} = \boldsymbol{dW}.\text{sum}(\dim = 0)$

27: Return $\mathbf{dQ}, \mathbf{dK}, \mathbf{dV}, \boldsymbol{dW}$.

---

## C. More case studies

*...An alternate approach is given below in Corollary \ref{cor:infty}.* Along the way we obtain more information about the eigenfunctions, which leads directly to an explicit formula for $u_m(x; \infty)$, see \eqref{conjsum2} and \eqref{Bkmexplicit}. As $\sigma$ increases, the derivatives of $u_m(x; \sigma)$ remain bounded, and so to ensure that the interior condition in \eqref{deltabc} continues to hold, the values $u_m(x_k; \sigma)$ *converges to infinity....*

*...must converge to* 0 *as $\sigma$ converges to infinity.* Our first corollary of Theorem \ref{thm:main} is that these values converge to 0 at the same rate for each node $x_k$. \begin{cor} \label{cor:nodes}. Up to an *overall normalization factor, for each $\sigma \geq 0$ and $1 \leq m \leq n-1$, the values of the eigenfunctions...*

*...To obtain the limiting eigenfunctions, which we denote by* \label{def:uminfty} $\nu_m(x; \infty) = \lim_{\sigma \to \infty} u_m(x; \sigma)$, one can use the fact that $\gamma_m(\sigma) \to n\pi$ for $1 \leq m \leq n$ *to obtain...*

*...the eigenvalues $la_m(\sigma)$ for $1 \leq m \leq n$ all converge to $la_n = n^2\pi^2$ as $\sigma$ tends* **to infinity. (Note that this is consistent with our implicit expression for the eigenvalues $la_m(\sigma)$ from Theorem \ref{thm:main}.) From Corollary \ref{cor:nodes}**, this ensures that $u_m(x_k; \sigma)$ converges to zero as $\sigma$ tends to infinity. This means that $u_m(x; \infty)$ (defined in \eqref{def:uminfty}) is proportional...

Figure 5: In the case above, retrieved top-1 chunk introduces the definition used in the target chunk, while the adjacent retrieved 3rd best chunk and retrieved 4th best chunk both cover the same variants as those appear in the target chunk. Retrieved 2nd best chunk contains the theorem and corollary used in the **query chunk**.

*... denote the projection $\pi : \rpvc \to \cpvc.$*\begin{lemma} \label{lemma:mu circ pi=2n} *The complex and real moment maps for $G^{\mathbf{C}}$ are related by $\mu^* \circ \pi = 2n$*\end{lemma}\begin{proof} *Many of our computations...*

*... $\pi(\omega([v])) = \omega(\pi[v])$* **where $\omega(p)$ denotes the $\omega$-limit set of the negative gradient flow starting from $p.$\end{prop}\begin{proof} Applying Lemma** \ref{lemma:mu circ pi=2n} we have $4 < grad||n||^2[v], w_{[v]} >= 4$ ...

Figure 6: In the case above, retrieved top-1 chunk introduces the lemma used in the target chunk.

*... They are not the same: see Section \ref{sec:15}. To establish Theorem \ref{thm:ABn} it suffices to prove it for $B(n)$; the estimate for $A(n)$ then follows from the linear relation $A(n) = \log G_n + B(n)$(from* \eqref{eqn:GABx} *) combined with the asymptotic estimate for $G(n)$ in* \eqref{eqn:logG-asymp}. *The main contributionin the sum $B(n)$ comes from those primes $p$ having $p > \sqrt{n}$, whose key property ...*

*... that exponential sum methods yield alternative unconditional* estimates for $A(n,x)$, $B(n,x)$ and $\log G(n,x)$ , which are nontrivial when $x = o(n)$, and apply for $x > \sqrt{n}$. These estimates improve *on the estimates of our main theorems for certain ranges ...*

*...* exponents $\nu_p(G_n)$ as a difference of quantities given by statistics of the base $p$ radix expansion of integers up to $n$ (see Theorem \ref{thm:explicit} ). Summing over $p \leq x$ yields a formula $\log G(n,x) = A(n,x) - B(n,x)$ involving nonnegative arithmetic *functions $A(n,x)$ and $B(n,x)$ ...*

**... *The implied constant* in the $O$-notation does not depend on $\alpha$. \end{thm} The limit function $f_B(\alpha)$ is pictured in Figure \ref{fig:B2}. The function lies strictly above the diagonal line** $\beta = (1 - \gamma)\alpha$; note that in \eqref{eqn:GABx} in its relation to $\log G(n,x)$ it appears with a negative sign, consistent with $f_G(\alpha)$ ...

Figure 7: In the case above, retrieved top-1 chunk and retrieved 3rd best chunk are adjacent, which mentions the same equation as target chunk. retrieved 2nd best chunk and retrieved 4th best chunk both mention $\log G(n,x)$ , which also appears in target chunk.

## D. The details for the NIAH test

In all evaluations conducted for the NIAH tests, we fine-tune all models using checkpoints derived from PG19, employing the same set of synthetic data. The number of fine-tuning steps is set to one-tenth of the total steps used during pre-training, while all other hyperparameters are kept constant. Examples of the synthetic data utilized for each task are presented in the table 5. Specifically, we pad the input tokens to ensure that the landmark token can be inserted before "is" in the question.

| Task | Example |
|------|---------|
| Single NIAH | (essays)... 
 The passkey is: {tokens}. 
 ... 
 What is the passkey? The passkey is {tokens}. |
| 2-hop NIAH with noises | (essays)... 
 DEF {tokens_5}->{tokens_6} ... 
 DEF {tokens_2}->{tokens_3} ... 
 DEF {tokens_4}->{tokens_5} ... 
 DEF {tokens_1}->{tokens_2} ... 
 ... 
 The path from {tokens_1} is: {tokens_2}, {tokens_3} |

Table 5: Task examples for the two NIAH tests.

## E. Discussions about the RPT_{Contriver} baseline

In the original RPT, a reference LM is used to pre-prepare target chunks for each chunk, as discussed in the related works. Compared to using Contriever as the retrieval module, the original method offers stronger causal retrieval capabilities. However, since the code for retriever distillation in the original RPT is not released and the approach is costly and less flexible, we opt to use Contriever instead. $RPT_{contriever}$ can be considered a fusion of RETRO and RPT. It retrieves past chunks in a manner similar to RETRO and integrates the retrieved information in the style of RPT. The retrieval process involves dividing every 64 tokens into a chunk, encoding them with Contriever to obtain chunk representations, and then retrieving past chunks based on cosine similarity with the current chunk.

