# OpenReview forum: "Efficient Length-Generalizable Attention via Causal Retrieval for Long-Context Language Modeling"
_ICML.cc/2025/Conference — ICML 2025 poster_

### Official Review · Reviewer_PLe3 · 2025-03-04

**Overall Recommendation:** 2

**Summary:**

This paper proposes an attention mechanism, Grouped Cross-Attention (GCA), to improve long-context language modeling. By integrating a retrieval mechanism directly into the attention computation, GCA allows Transformers to generalize to significantly longer contexts while maintaining computational efficiency. The authors also introduce the Differentiable Retrieval-based Transformer (DRT), which utilizes GCA to retrieve and integrate relevant past chunks dynamically. Experimental results demonstrate that DRT achieves superior performance in long-context modeling tasks.

**Claims And Evidence:**

While the paper presents some evidence for its claims, these significant gaps in explanation, methodology, and evaluation make it difficult to conclude that the claims are fully supported by clear and convincing evidence.

**Essential References Not Discussed:**

The reference is good.

**Experimental Designs Or Analyses:**

There are significant concerns about the experimental methodology, particularly regarding parameter justification, comparison fairness, and incomplete exploration of model variations. These issues potentially undermine the strength and generalizability of the paper's findings, suggesting that while the core approach may be promising, the experimental validation falls short of convincingly demonstrating its effectiveness.

**Methods And Evaluation Criteria:**

While the proposed method addresses a relevant problem and uses some standard evaluation approaches, there are significant gaps in the methodological clarity and evaluation comprehensiveness that limit the conclusiveness of the paper's findings. The evaluation would be stronger with more diverse metrics, standard benchmarks, and clearer explanation of parameter choices and performance variations across datasets.

**Other Comments Or Suggestions:**

1. Figure 2 is very confusing. Based on the figure, it appears that chunk c_6 retrieves the top-k relevant past chunks for its next chunk using only landmark representation l_6 and previous chunks’ landmark representations encoded by the bi-directional Transformer encoder. However, according to Equation (1), the causal relevance score is computed using the landmark representation output by the previous decoder layer of the g-th group. The concept of grouping (g-th group) is only introduced in the upper decoder. This discrepancy makes Figure 2 very misleading, making the whole methodology difficult to understand. I strongly suggest that the authors provide further clarification on this aspect and improve the figure to enhance readability.

**Other Strengths And Weaknesses:**

Strength
1. The paper introduces a useful and effective method for handling long-context sequences, which is a crucial problem in Transformer-based language modeling.
2. GCA incorporates the relevance scores of retrieval chunks into the LM decoder, allowing the retriever to adaptively learn how to retrieve through training for predicting subsequent tokens. This manner for retrieval-based language models is novel and reasonable.
﻿
Weakness：
1.Some annotations and explanations are not sufficiently clear, making the proposed method difficult to understand. See Questions 1, 2, and 3 for details.
2.The notation in the equations is inconsistent. For instance, the representation of chunk-wise CA outputs in Figure 2 caption does not match the notation used in Equation (2). Please unify the notation for clarity.
3. In practice, a chunk size of 64 and a retrieval number of 8 are chosen as parameters for model training in this paper. What is the rationale behind selecting these specific parameters? Are these preferred settings consistent across different tasks? Are there any ablation studies that address this issue?
4. In Table 1, there is a noticeable difference in perplexity between the PG19 dataset and the ArXiv-math dataset. What accounts for this discrepancy?
5. Since the GCA method integrates retriever into the training process of predicting subsequent tokens, comparing its perplexity with baseline methods might be unfair. It would be better to test and compare on more practical tasks (such as LongBench benchmark).

**Questions For Authors:**

1. Line 163: Could you provide a more detailed explanation of C_k and l_k? The paper only gives a vague description. Based on my understanding: C_k represents the token representations of chunk k, encoded by the bi-directional Transformer encoder. L_k is the landmark representation of chunk k, which (if I haven’t missed anything) is only briefly mentioned in Figure 2 caption. Could you clarify if this interpretation is correct?
2. Equation (2): What normalization method does Norm() represent? The paper does not specify this.
3. Line 324: What is the "off-the-shelf retriever" used by the authors? Please specify.
4. The paper only tests cases where the group number (G) is 1 and 2. Have you experimented with a wider range of group numbers? If so, what is their impact on performance?
5. Figure 3: Which variant of the model is used here—DRTretrieval×1 or DRTretrieval×2? Please clarify.

**Relation To Broader Scientific Literature:**

The paper tackles the important challenge of extending Transformer models to handle longer contexts, which is a recognized limitation of standard attention mechanisms. This connects to the broader literature on efficient Transformers and long-context modeling.

**Theoretical Claims:**

There don't appear to be explicit theoretical proofs or claims in the paper. The paper primarily focused on introducing a new attention mechanism (GCA) and a model architecture (DRT) with empirical evaluations rather than presenting mathematical proofs or theoretical guarantees.

---

> ### Author Rebuttal · Authors · 2025-03-27
>
> Thank you very much for reviewing our manuscript.
>
> **W2. For instance, the representation of chunk-wise CA outputs in Figure 2's caption does not match the notation used in Equation (2).**
>
> If the inconsistency refers to the subscripts, it can be easily fixed by rewriting $O_{t+1,k}^l$ to $O_{t+1}^{l,k}$. Besides, eq(2) is the batch version of Figure 2.
>
> **Cmt 1. Figure 2 is very confusing..**
>
> Thanks for your comment.
>
> In fact, using landmarks for retrieval can be regarded as a special case when group = 1. Note that $l_k$ is essentially $h_k^0$ passed through the encoder.
> To make it easier to understand, we can additionally explain when G=1,  $r_t^k=l_t^\top l_k / \sqrt{d}$. As the retrieval is conducted only once and does not involve the use of landmark representations from different layers.
>
> **W3. Regarding parameter justification**
>
> As hyperparameters cannot be exhaustively studied, we follow the setups from previous works ([1][2][3]), where a chunk size of 64 is commonly chosen, along with a 2^n chunk number. We set the chunk number to 8 for better alignment with landmark attention and sliding window, which can theoretically make the attention fields consistent as described in lines 312-319.
>
> **W4. Regarding the difference in perplexity between the PG19 dataset and the ArXiv-math**
>
> Our results are consistent with all previous paper works ([1][2][4][5]).
> The ArXiv-math dataset is a collection of mathematical papers that involve a significant amount of logical symbolic reasoning. The patterns within this dataset are more structured and regular, leading to a lower perplexity.
>
> **W5(a) Since the GCA method integrates retrieval, comparing its perplexity with baseline methods might be unfair.**
>
> Please note that our baselines, like RPT and Landmark Attention, also integrate retrievers into the training process, while Block Recurrent Transformer employs recurrent mechanisms to maintain long-range memory. They are designed for long-context. Thus, it's not unfair to compare with them.
>
> Moreover, we also evaluate downstream task performance in Table 2.
>
> **W5(b) It would be better to test and compare on more practical tasks (such as LongBench benchmark).**
>
> We believe that LongBench is not well-suited for evaluating small models.
> In fact, most results of small models on LongBench are even inferior to random guessing among four options. Here we report results on LongBench v2 for reference, following the evaluation method used in the cloze task presented in this work (https://arxiv.org/abs/2406.07887):
>
> |Model|Overall | Easy | Hard | Short | Medium | Long |
> |-|-|-|-|-|-|-|
> |TFM /sw | 24.0 | 22.9 | 23.2 | 27.2 | 20.0 | 26.9 |
> |Block Recurrent TFM | 22.8 | 20.8 | 25.1 | 19.4 | 22.8 | 25.9 |
> |Landmark | 24.3 | 23.4 | 25.7 | 28.3 | 19.1 | 25.0 |
> |RPT | 23.3 | 25.5 | 22.2 | 21.1 | 25.6 | 22.2 |
> |DRTretrieval×1 | 24.8 | 28.6 | 23.8 | 21.1 | 22.8 | 27.8 |
> |DRTretrieval×2| 25.6 | 24.0 | 25.7 | 27.2 | 25.1 | 25.9 |
>
> **Q1. Could you provide a more detailed explanation of C_k and l_k?**
>
> To elaborate, each chunk contains S tokens and 1 landmark token. These tokens are fed into the bi-encoder, where the representations of the S tokens correspond to $C_k$, while the representation of the single landmark token corresponds to $l_k$.
>
> **Q2. Equation (2): What normalization method does Norm() represent?**
>
> Both RMSNorm and LayerNorm are valid options. In our implementation, DRT is built upon LLama, where RMSNorm is employed.
>
> **Q3: Line 324: What is the "off-the-shelf retriever" used by the authors?**
>
> At lines 323–324, it states: "w/ contriever: We replace the relevance scores in GCA by using an off-the-shelf retriever." From this, the "off-the-shelf retriever" refers to Contriever.
>
> **Q4: The paper only tests cases where the group number (G) is 1 and 2. ... what is their impact on performance?**
>
> Before applying the softmax off-by-one in Equation(3), we observed that increasing the number of groups resulted in a slight increase in perplexity. However, after applying the softmax off-by-one, PPL generally decreases as the number of groups increases. We suppose that softmax off-by-one enables GCA to disregard noisy retrieved chunks, thereby enhancing performance.
>
> **Q5: Figure 3: Which variant of the model is used here—DRTretrieval×1 or DRTretrieval×2?**
>
> We use DRTretrieval×1 by default for all experiments.
>
> References:
>
> [1] LANDMARK https://arxiv.org/abs/2305.16300,
>
> [2] RPT https://aclanthology.org/2024.tacl-1.66/,
>
> [3] RETRO https://arxiv.org/abs/2112.04426
>
> [4] BRT https://arxiv.org/abs/2203.07852,
>
> [5] cepe https://arxiv.org/abs/2402.16617,
>
>
> In summary, all clarity issues can be fixed with minor adjustments. We follow previous work for choosing hyperparameters, fairly compare with competitive baselines, and use standard evaluation metrics in the long-context area. We hope our replies address your concerns. We would greatly appreciate it if you could consider increasing the score.

---

### Official Review · Reviewer_wqhZ · 2025-03-14

**Overall Recommendation:** 3

**Summary:**

The paper introduces a new attention mechanism to integrate dynamic context called Grouped cross attention (GCA). GCA helps maintaining long term dependencies during sequence generation enabling long range information access and length generalization. GCA integrates chunk to chunk retrieval to learn and retrieve past chunks to reduce autoregressive generation loss for next time step prediction in the decoding process. Based on GCA in transformers layers, the authors introduce differentiable retrieval based transformers (DRT) to enable pretraining across longer sequences and discuss hardware aware implementation. DRT showcases performance gains over very long range sequence generation via empirical evaluation.

**Claims And Evidence:**

yes

**Essential References Not Discussed:**

N/A

**Experimental Designs Or Analyses:**

DRT is compared to SOTA literature in the field and showcases marginal performance gains compared to existing literature across various datasets and sequence generation tasks. Ablation studies are performed to strengthen the claims.

**Methods And Evaluation Criteria:**

The work proposes the DRT model architecture using GCA attention mechanism. Experiments are conducted to compare DRT on various datasets and tasks and comparisons are made w.r.t SOTA literature in the field. Standard evaluation metrics corresponding to each dataset/task is used.

**Other Comments Or Suggestions:**

N/A

**Other Strengths And Weaknesses:**

N/A

**Questions For Authors:**

N/A

**Relation To Broader Scientific Literature:**

The authors present a novel attention mechanism to preserved information across long range sequences and improve language modeling empirical performance across various sequence generation tasks. The broader scientific community would benefit from the results and model architecture presented by the study.

**Theoretical Claims:**

No

---

> ### Author Rebuttal · Authors · 2025-03-27
>
> Thank you very much for your review comments and support for this work!
>
> Although the improvement on perplexity is relatively marginal, the performance of passkey retrieval in Table 2 is quite significant, especially when the context length far exceeds that of the pre-training stage. This demonstrates extraordinary extrapolation capabilities, which none of the existing attention mechanisms possess.

---

### Official Review · Reviewer_gMz8 · 2025-03-14

**Overall Recommendation:** 4

**Summary:**

This paper introduces grouped cross attention (GCA), where the model learns to retrieve past chunks of tokens to reduce the prediction error on future tokens.  The model is trained end-to-end to retrieve relevant chunks, and thus does not depend on a fixed retriever.  The GCA modules are appended after attention modules in a transformer architecture.  They offload the hidden states of past chunks to CPU to save memory, which enables them to significantly reduce memory overhead relative to competing methods and enable large contexts.  Their throughput is similar to a simple sliding window attention baseline.  They evaluate, PG19, arXiv math, single key NIAH, and summarization tasks.  They compare against baselines RPT, landmark attention, and Block recurrent TFM.

## Update after rebuttal

The authors addressed my concerns during the rebuttal stage.  I maintain my score and positive assessment of the paper.

**Claims And Evidence:**

Claims:
* "GCA is the first attention mechanism that can achieve perfect passkey retrieval with 16M context length, 1000× the pre-training length"
* "DRT significantly outperforms all baselines with comparable pre-training costs and much lower
inference costs"
* In discussion: "GCA can extrapolate over 1000 times with stable PPL and perfect passkey retrieval, but Landmark Attn
fails".

Evidence:
* This claim is verified in Table 2 where they get 100% on single NIAH at 16M context length.
* The perplexity evals are verified in Table 1 where they beat the baselines by a small margin for the large contexts.  In Table 2 they significantly beat the baselines on NIAH.  The throughput and memory overhead is documented in Figure 3 and Table 3.  Table 3 demonstrates they have significantly smaller memory overhead compared to landmark attention and have similar throughput to the sliding window baseline.  It would help to show the throughput and memory overhead of the other baselines RPT, and block recurrent TFM.
* Extrapolation to 1000x is tested on NIAH but not on PPL (which is only evaluated up to 32K), can you please clarify or correct this?

**Essential References Not Discussed:**

They seemed to have missed the reference [1], which is similar to Transformer-XL in that they segment the input into chunks and maintain summary tokens to summarize prior tokens.  I will also mention concurrent works [2], [3] for the authors benefit to add to the camera-ready, but I do not expect comparisons against these works as they are concurrent so I am not penalizing the authors for this in the review.  [2] appeared within 1 month of the submission deadline and [3] appeared after the submission deadline.  [2], [3], summarize prior chunks into summary tokens which are used to retrieve chunks which are then used for cross-attention.  [3] notably also introduces optimized kernels, similar to the present submission.

[1] Tsendsuren Munkhdalai, Manaal Faruqui, and Siddharth
Gopal. Leave no context behind: Efficient infinite context transformers with infini-attention. arXiv preprint
arXiv:2404.07143, 2024

[2] Elvis Nunez, Luca Zancato, Benjamin Bowman, Aditya Golatkar, Wei Xia, and Stefano Soatto. Expansion Span: Combining Fading Memory and Retrieval in Hybrid State Space Models. arXiv:2412.13328, 2024

[3] Jingyang Yuan, Huazuo Gao, Damai Dai, Junyu Luo, Liang Zhao, Zhengyan Zhang, Zhenda Xie, Y. X. Wei, Lean Wang, Zhiping Xiao, Yuqing Wang, Chong Ruan, Ming Zhang, Wenfeng Liang, and Wangding Zeng. Native Sparse Attention: Hardware-Aligned and Natively Trainable Sparse Attention. arXiv preprint arXiv:2502.11089, 2025.

**Experimental Designs Or Analyses:**

They train 128M and 350M TinyLlama based models on 32B tokens, which is past Chinchilla optimal and is thus reasonable.  It would be helpful to show results with larger models.  While training larger models on 32B tokens may take too long with your 8 A100 setup, you could test throughput, memory overhead and plot scaling curves for the initial set of tokens for larger models to show that your method is scalable.

**Methods And Evaluation Criteria:**

The evaluation on RULER is pertinent for long-context, and language modeling on PG19 is standard for such works.

It would help to see evals on the full RULER suite including multi-key retrieval, as they only evaluate on single NIAH.

**Other Comments Or Suggestions:**

* It would be of great help to the community if you could provide the Triton code in the camera ready to supplement the pseudocode.
* Can you report throughput, memory overhead, and scaling curves for larger models (even if you don't train on the full 32B tokens)

Suggested edits:

Line 212: "the K, V **liner** transformations" --> "the K, V **linear** transformations"

Line 402: "confirming our hypothesis that conducting causal retrieval every **G-layer** contributes to scenarios requiring multiple retrievals" --> "confirming our hypothesis that conducting causal retrieval every **G layers** contributes to scenarios requiring multiple retrievals"

**Other Strengths And Weaknesses:**

Strengths:
* They achieve perfect recall on single NIAH up to 16M context length, while achieving reasonable perplexity up to 2x the pre-training context length
* They provide hardware-aware psuedocode for their Triton implementation of FlashGCA forward and backward
* The case studies on retrieving relevant llemas and definitions for math theorems on arXiv math is quite interesting and illuminating.

Weaknesses:
* They only evaluate on single NIAH
* The models are still relatively small 128M and 350M.  It would help to have some experiments with larger models.  The authors state that they trained with 8 A100 GPUs in bf16 precision.  In bf16 with Adam you should be able to train up to around a 3B model without model parallel assuming 40GB of HBM for an A100.  Although, it may take a long time to train on the full 32B tokens.  Perhaps some smaller scale fine-tuning experiments or other simplifications could be made to get signal at larger scale.  In any case, you don't need a large number of tokens in order to time the throughput and calculate the memory overhead relative to other methods.  This should be included in the camera-ready.
* Larger models will take up more GPU memory, so the 16M context result will likely not apply (as GPU memory will be exhausted quicker by the KV cache), however it is still a great proof-of-concept
* They are lacking throughput and memory calculations for the other baselines RPT, Block recurrent TFM.  Adding some additional curves to Figure 3a would be helpful.

**Questions For Authors:**

* It seems you only try setting the number of groups to 1 or 2.  Can you explain how you might select the groups for larger models and how that would affect your results?
* How would your method scale to LLMs with billions of parameters?  Can you provide a generic analysis of the time to offload the chunks to CPU based on the GPU to CPU communication speed to demonstrate scalability?
* Can you provide throughput, memory overhead, and scaling curves for larger models?

**Relation To Broader Scientific Literature:**

This work relates to broader literature on chunk retrieval and long-context language modeling.  They position themselves among related works such as RETRO, Landmark attention, RPT, Transformer-XL, Infini-attention,

**Theoretical Claims:**

This is an empirical work, there are no theoretical claims to justify.

---

> ### Author Rebuttal · Authors · 2025-03-27
>
> Thank you for your professional and constructive review.
>
> **W1. They only evaluate on single NIAH/It would help to see evals on the full RULER suite including multi-key retrieval,**
>
> Besides the single NIAH, we also evaluated the variable tracking task in Table 2, which is the most challenging and comprehensive among RULER NIAH tasks. Our setup includes 6 variables and 2-hop assignments, covering both multi-key and multi-hop scenarios. Results confirm that our length generalization capability remains effective even under these complex conditions.
>
> In addition, we are also willing to report the results of multi-key passkey retrieval, which contains 12 keys.
> | | 64K | 256K | 1M | 4M
> |-|-|-|-|-|
> TRMsw | 1.2 | 0.0 | - | - |
> LMK | 0.0 | - | - | - |
> DRTretreivalx1| 85.71 | 88.24 | 87.5 | 87.50
>
> **Q3//W2/W4. Can you report throughput, memory overhead, and scaling curves for larger models.**
>
> Sure, here are the results:
>
> | Model | Throughput (350M, 32K) | Memory（350M) | Throughput (760M, 16K) | Memory(760M)  | Throughput (1.5B, 8K)     | Memory(1.5B)  | Throughput (3B, 4K)     | Memory (3B)     |
> |-|-|-|-|-|-|-|-|-|
> | Transformers$_\text{SW}$  | 2.91e5 tokens/s | 50G | 1.31e5 tokens/s | 50G | 6.56e4 tokens/s | 50G  | 3.41e4 tokens/s | 72G  |
> | Transformers$_\text{fullattn}$| 4.44e4 tokens/s | 50G | 4.00e4 tokens/s | 50G | 3.50e4 tokens/s | 50G | 2.85e4 tokens/s  | 72G  |
> | BRT | 7.94e4 tokens/s | 59G  | 5.70e4 tokens/s | 56G  | 4.02e4 tokens/s | 59G | -  | OOM |
> | RPT$_{\text{Our impl}}$| 1.13e4 tokens/s | 57G | 5.17e4 tokens/s | 59G | 2.73e4 tokens/s | 64G | 1.45e4 tokens/s | 80G |
> | DRT$retrieval \times 1$| 2.32e5 tokens/s  | 56G | 1.06e5 tokens/s | 59G | 5.52e4 tokens/s | 63G | 2.93e4 tokens/s | 80G |
> | DRT$retrieval \times 2$| 2.28e5 tokens/s  | 57G | 1.05e5 tokens/s | 59G | 5.52e4 tokens/s | 63G  | 2.93e4 tokens/s | 81G |
>
> Here, we did not enable FSDP or gradient checkpointing. If enabled, the 3B model should achieve over 8K context length.
> Compared to full attention, our approach shows significant speed advantages at 8K and above. Compared to sliding window, the additional overhead is around 20\%, but while gaining the ability to capture ultra-long-range information. Overall, DRT is scalable.
>
> If we have the opportunity to submit a camera-ready version, we will include the corresponding scaling curve for this data.
>
> **Q1. Can you explain how you might select the groups for larger models and how that would affect your results?**
>
> Before applying the softmax off-by-one in Equation(3), we observed that increasing the number of groups resulted in a slight increase in perplexity. However, after applying the softmax off-by-one, PPL generally decreases as the number of groups increases. We suppose that softmax off-by-one enables GCA to disregard noisy retrieved chunks, thereby enhancing performance.
> Increasing groups essentially expands the attention fields because each retrieval may access different chunks. However, the marginal gain diminishes, and the memory footprint also rises with the increase in the groups. Therefore, for larger models, we would likely still set the group number between 1 and 2, balancing computational cost and performance.
>
> **Q2. Can you provide a generic analysis of the time to offload the chunks to CPU based on the GPU to CPU communication speed to demonstrate scalability?**
>
> The offloading time from GPU to CPU is proportional to the KV cache size, which is determined by (batch_size, chunk_num, chunk_size, hidden_size, groups). Note that this is independent of the number of layers, as the gca KV is shared within the same group.
>
> Assuming a large model with a hidden size of 4096 and bfloat16 precision (2 bytes per value), for batch size = 1, group = 1, chunk size = 64, and input length = 1M tokens, the total KV cache size is:
> 1 * (1M // 64) * 64 * 4096 * 1 * 2(bf16) = 8192 MB = 8 GB
> With typical GPU-to-CPU communication speeds exceeding 20 GB/s, all required KV caches for GCA can be offloaded in less than 0.4 seconds, which is fully scalable.
>
> **Missing Reference:**
>
> If we have the opportunity, we will include these references in the camera-ready version.
>
> By the way, NSA was not released on arXiv at the time of our submission. During the review period, we evaluated its extrapolation capabilities for the S-NIAH task based on (https://github.com/fla-org/native-sparse-attention) and found that its extrapolation ability is still limited. The results are as follows:
>
> | | 16K | 64K | 256K | 1M
> |-|-|-|-|-|
> |NSA | 98.3 | 50.8 | 18.4 | oom |
>
> This result shows that GCA still exhibits significant advantages in terms of length generalization.
>
> Thank you once again for your thoughtful and professional review. Your comments have been invaluable in helping us refine our work and improve our paper.
> We sincerely hope that our responses have addressed your concerns, and we would be truly grateful if you could kindly offer us more support.

---

> > ### Comment · Reviewer_gMz8 · 2025-04-03
> >
> > I thank the authors for their response.  I had overlooked the results on variable tracking on RULER, thank you for pointing that out.  I think the throughput benchmarking is convincing, as the throughput is about 80-85% of sliding window attention (not bad), and beats the other baselines.  I think these additions would greatly improve the paper and increase its impact.  I have updated my score from 3 to 4 based on this information.

---

> > > ### Author Response · Authors · 2025-04-03
> > >
> > > We sincerely thank you once again for the thoughtful and insightful suggestion! Evaluating throughput is a great idea to showcase scalability of DRT.

---

### Decision · Program_Chairs · 2025-05-01

**Decision:**

Accept (poster)

**Comment:**

This paper proposes Grouped Cross Attention (GCA), a novel attention mechanism that enables Transformers to generalize to context lengths 1000× longer than those seen during training while maintaining a constant attention window size. By learning to retrieve the most relevant past chunks end-to-end, GCA significantly reduces memory and computational costs while achieving strong long-range information access and near-perfect performance on passkey retrieval tasks with up to 16M token contexts.

In general, the idea of the paper is interesting and the experimental results support the claim. Nonetheless, the reviewers have a few concerns about the clarity of the proposed method’s presentation, the limited evaluation on small models and a single dataset, missing comparisons with similar or concurrent works, lack of throughput and memory benchmarks against baselines, and the absence of ablation studies and justification for key hyperparameter choices. During the rebuttal, the authors attempted to address the reviewers’ questions, but their responses, particularly to reviewer PLe3, did not fully resolve the concerns.